# Re-Evaluating the Treatment Plan for Diabetic Macular Edema Based on Early Identification of Response and Possible Biochemical Predictors of Non-Response After the First Intravitreal Ranibizumab Injection

**DOI:** 10.3390/biomedicines13102438

**Published:** 2025-10-07

**Authors:** Sameh Mohamed Elgouhary, Noha Rabie Bayomy, Mohamed Khaled Elfarash, Sara Zakaria Aboali, Sara Abdelmageed Barakat, Mona Abdelhamid Elnaggar, Noha Khirat Gaber

**Affiliations:** 1Department of Ophthalmology, Faculty of Medicine, Menoufia University, Shebin Elkom 32511, Egypt; noha.khirat1985@gmail.com; 2Department of Medical Biochemistry and Molecular Biology, Faculty of Medicine, Menoufia University, Shebin Elkom 32511, Egypt; noharabie@yahoo.com; 3Department of Medical Biochemistry and Molecular Biology, Faculty of Medicine, Benha National University, El Obour City 13518, Egypt; 4Tanta Ophthalmology Hospital, Ministry of Health and Population, Cairo 32111, Egypt; mkh701024@gmail.com (M.K.E.); sarazakaria1512@gmail.com (S.Z.A.); 5Shebin Elkom Ophthalmology Hospital, Ministry of Health and Population, Cairo 32111, Egypt; sarabarakat01@gmail.com; 6Samannoud Central Hospital, Ministry of Health and Population, Cairo 32111, Egypt; monaabdelhamid131@gmail.com

**Keywords:** diabetic macular edema, real-time PCR, Ranibizumab, response, anti-VEGF

## Abstract

**Background**: This study aimed to change the current concept of diabetic macular edema (DME) management through (1) early categorization of our DME patients into either responders or non-responders after the first intravitreal Ranibizumab (IVR) injection, and (2) finding a suitable clinical–biochemical diagnostic panel to identify the possible cause(s) of non-response in each non-responder and changing the treatment plan in each particular patient accordingly. **Patients and methods**: Our study included 64 eyes of 40 patients with DME **(Group A, DME patients)** and 40 eyes of 40 healthy individuals matched for age and sex **(Group B, controls)**. Blood and aqueous samples were collected from the study participants before and one month after IVR injection. The DME patients were further subdivided into responders and non-responders according to their response to the first IVR injection. Lymphocyte activation markers, NETosis markers, angiogenic factors, astrocytes, innate immunity, and inflammasome markers were assessed in both groups. **Results**: Multivariate regression analysis revealed that macular ischemia, aqueous levels of *hexokinase 1*, *SELL CD62L*, *ELANE*, *MPO*, *VEGFA*, *and SEMA4D* were the most significant factors affecting the response to IVR (*p* < 0.05). **Conclusions**: defining our DME patients as responders and non-responders after the first IVR injection, combined with potential utilization of a clinical–biochemical panel (macular ischemia- PCR array of combined *Hexokinase 1*, *MPO*, and *SEMA4D*) in each non-responder, may represent a good starting point for changing the current DME management strategy.

## 1. Introduction

Diabetic retinopathy (DR) is one of the most common causes of legal blindness worldwide [1,2]. Diabetic macular edema (DME) is the most significant cause of central vision loss in DR patients. The risks for DME are duration and control of diabetes (represented by levels of glycosylated hemoglobin (HbA1c) [3].

Diabetic macular edema results from blood–retina–barrier disruption due to chronic hyperglycemia and alterations in several biochemical pathways, leading to vascular leakage, fluid accumulation, and macular thickening. However, the pathogenesis of DME has not yet been fully clarified. Elevated vitreous levels of vascular endothelial growth factor (VEGF) with increasing vascular permeability are known to play a role in the development of DME [4,5].

Ranibizumab (Lucentis^®^, Genentech, South San Francisco, CA, USA/Roche, Basel, Switzerland) is designed for ocular use against *VEGF-A*. Ranibizumab, in 2012, was the first approved drug for DME. Its approval for DME was based on the results of the RISE and RIDE phase 3 clinical trials [6]. The DRCRNet protocol S concluded that Ranibizumab was non-inferior to retinal laser photocoagulation regarding visual improvement and visual field changes in diabetic patients with proliferative DR [7,8]. However, some studies have shown that some patients still respond poorly to IVR therapy, even after three or more injections [9]. This resistance may be due to other interfering factors in DME pathogenesis.

NETosis markers, NETs referred to as neutrophil extracellular traps, are formed when neutrophils release their DNA and granular contents, which are involved in the pathogenesis of various inflammatory diseases, including DME. NETosis biomarkers such as myeloperoxidase (*MPO*) and protein arginine deiminase 4 (*PAD4*) show elevated levels in the serum of diabetic patients [10]. 

Recent studies have discussed the role of autoimmunity in chorioretinal vascular disorders. The inner (retinal vascular endothelial cells) and outer (tight junctions between the retinal pigment epithelial (RPE) cells) blood–retinal barriers (BRB) sequester retinal auto-antigens. In DR, vascular hyperpermeability, or the disruption of outer or inner retinal–blood barriers, allows immunological agents to react with sequestered antigens, leading to autoantibody formation against retinal antigens and activation of innate immunity [11,12]. Thus, these antibodies can be used as biomarkers for DR and DME [13].

The imbalance between proangiogenic and antiangiogenic factors, favoring the former with higher levels of VEGF and angiopoietin-2 (*Ang-2*), plays a vital role in the pathogenesis of DR and DME [14].

Persistent macular edema for more than a month can lead to permanent damage to retinal cells, causing irreversible vision loss [15]; in addition to the fact that the earlier the management of any disease, the better the prognosis, this urged us to the first aim of our trial, to change the current DME treatment paradigm by earlier definition of our DME patients into responders and non-responders after the first intravitreal injection of anti-VEGF (instead of the current plan of waiting for 3–5 monthly loading doses to assess the patients’ responses). In medicine, we have a golden rule: “Successful treatment of a disease depends on hitting its exact pathophysiology.” From this perspective, we sought to identify a suitable clinical–biochemical diagnostic panel to determine the possible cause(s) of non-response in each non-responder, thereby adjusting the treatment plan for each patient accordingly, which was the second aim of our study.

## 2. Patients and Methods

This prospective and interventional study was conducted at the Ophthalmology and Medical Biochemistry departments of the Faculty of Medicine at Menoufia University between April 2022 and September 2023. The laboratory work was conducted in the Central Laboratory of the Faculty of Medicine, Menoufia University. This study included 80 subjects classified into two groups (the selection of participants followed a simple random technique using computer-generated sequences to assign participants to study groups, ensuring that both groups were comparable and minimizing selection bias by chance. Then, participants’ allocation to the study groups was concealed and blinded.

**Group A** included 40 diabetic patients (64 eyes) with DME requiring intravitreal injection of anti-VEGF Ranibizumab with the following criteria: age 18 years or older, type I or II diabetes mellitus (DM), diabetic macular edema causing visual loss with BCVA between 0.6 and 1.1 Log MAR, and central macular thickness (CMT) of 300 microns or more measured by optical coherence tomography (OCT). Patients with the following criteria were excluded from the study: history of vitreoretinal surgery, history of previous intraocular injection of corticosteroids or anti-VEGF, macular grid or micro-pulse laser treatment, the presence of significant media opacity that would limit vision recovery (e.g., dense cataract, vitreous haemorrhage), the presence of any retinal disease other than DR (e.g., macular degeneration, retinal vascular occlusions), co-existing vitreomacular traction or epiretinal membrane as determined by SD-OCT and other causes of macular edema (e.g., branch and central vein occlusion, retinal telangiectasia, and after cataract surgery), and patients with any disease causing elevated HbA1c other than diabetes, e.g., anemia, hemoglobinopathies, chronic liver disease, diabetic nephropathy, and renal failure.

**Group B consisted of** 40 healthy individuals (40 eyes), age- and sex-matched, who underwent cataract surgery.

### 2.1. The Ethical Committee Approval

The ethical committee approval of the Menoufia Faculty of Medicine and the respective approvals of the review board were obtained before proceeding with the study (trial registration number 3/2021 OPHT43). Additionally, this study adhered to the principles outlined in the Declaration of Helsinki. The study protocol, including its benefits and potential complications, was explained to all participants, and all recruited patients completed and signed the ‘informed consent’ form.

All patients with DME, before and one month after IVR injection, were subjected to complete ophthalmic examinations, including visual acuity examination using the Landolt C visual acuity chart, expressed in Log MAR for data analysis, measuring their refractive state using an auto refractometer (Topcon RM-8000B, Topcon Corporation, Tokyo, Japan), intraocular pressure (IOP) assessment using a Goldman applanation tonometer, slit-lamp examination, dilated fundoscopy using slit lamp and + 90D or + 78D lens, fundus fluorescein angiography (FFA) using the Topcon TRC-50 device (Topcon Corporation, Tokyo, Japan), and spectral-domain Optical Coherence Tomography (SD-OCT) using the Topcon 3D OCT-2000 (Topcon Corporation, Tokyo, Japan).

The patients with DME (Group A) were further subdivided into responders and non-responders. The response to the first IVR injection is defined as a reduction in CMT by more than 10% compared to baseline, using the ROC curve (anatomical response), and an improvement in BCVA by more than five ETDRS letters compared to baseline (functional response).

### 2.2. Intravitreal Injection of Ranibizumab

Ranibizumab (Lucentis; Novartis, Basel, Switzerland) 0.5 mg/0.05 mL (10 mg/mL) prefilled syringes were used for intravitreal injection. Topical anaesthesia was administered using benoxinate hydrochloride 0.4% eye drops. The procedure was performed under strict aseptic conditions, with the application of a sterile drape. A sterile eyelid speculum was used for proper eyelid retraction. A sterile calliper was used to measure the injection site 4 mm from the limbus for phakic patients and 3.5 mm for pseudophakic patients. Ranibizumab was injected inferotemporally, directed toward the vitreous center. A sterile cotton-tip applicator was applied to the injection site upon syringe withdrawal. Paracentesis was performed to withdraw approximately 0.05 mL of aqueous humor for analysis. Gatifloxacin 0.3% eye drops were prescribed four times daily for one week post-procedure. Patients were examined on day 1 and week 1 post-procedure. Another aqueous sample was collected at the time of the second injection (one month later) for testing the genetic biomarkers.

### 2.3. Sample Collection

Peripheral venous blood (5 mL) was collected from all the study populations, and 0.05 mL of aqueous humor samples were collected from 64 eyes in Group A and from 40 eyes in Group B. The samples were delivered immediately in a Vacutainer plastic tube containing EDTA for immediate RNA extraction using the miRNeasy Mini Kit (QIAGEN, Applied Biosystems, Foster City, CA, USA). The RNA extract was stored at −80 °C till analysis.

### 2.4. Assuring RNA Quality and Purity

The purity of RNA was determined by measuring its absorbance at 260 nm (A260) using a Nanophotometer N-605. The ratio between the absorbance values at 260 and 280 nm (A260/A280) estimates RNA purity. The 260/280 ratio of the selected RNA extract was between 1.8 and 2, which is acceptable.

The first step of real-time PCR (cDNA synthesis) was performed using the RevertAid First Strand cDNA Synthesis Kit from ThermoFisher Scientific Inc., Foster city, CA, USA, according to the manufacturer’s instructions.

The second step of real-time PCR (cDNA amplification with SYBR Green with low ROX) was performed with the QuantiTect SYBR Green PCR Kit (Applied Biosystems, USA) for the detection of

-**Lymphocyte activation markers:*** Hexokinase-1*, *Recoverin (RCVRN)*, *S100 calcium-binding protein A8 (S100A8) (Calprotectin)*, *and selectin L (SELL) (CD62L)* (**to examine the autoimmune element of DME pathology**).-**NETosis markers:*** Elastase neutrophil expressed (ELANE)*, *peptidyl arginine deiminase4 (PAD4)*, *and Myeloperoxidase (MPO)* (**to examine the inflammatory element of DME pathology**).-**Angiogenic biomarkers:*** VEGF-A*, *semaphorin 3A (SEMA3A)*, *semaphorin 4D (SEMA4D)*, *angiopoietin 1 (ANGPT1)*, *and angiopoietin 2 (ANGPT2)*.-**Astrocytic biomarker** (*Glial cell-derived neurotrophic factor (GDNF)*) (**to examine the vascular element of DME pathology**).-**Innate immunity and inflammasome markers:*** miR-135a-5p* (**mixed vascular and inflammatory**)* with thioredoxin-interacting protein (TXNIP)*, *NLR family pyrin domain containing 3 (NLRP3)*, *and receptor for advanced glycation end-products (RAGE)* (**to examine the inflammatory element of DME pathology**).

The human-tested genes were supplied by Thermo Fisher Scientific, Inc. Glyceraldehyde-3-phosphate dehydrogenase (GAPDH) was used as an internal reference gene. A final volume of 20 μL was used for each PCR reaction, containing 10 μL of SYBR Green 2× QuantiTect PCR Master Mix (Applied Biosystems, Foster City, CA, USA), 3 μL of cDNA, 1 μL of forward primer, 1 μL of reverse primer, and 5 μL of RNase-free H2O. The cycling conditions of the PCR reaction mix were as follows: incubation for 3 min at 94 °C, followed by 50 cycles, denaturation for 30 s at 94 °C, annealing for 40 s at 55 °C, and extension for 31 s at 72 °C. (Figure 1). Data analysis was performed with the Applied Biosystems 7500 software version 2.0.1.

The expression RQ (relative quantification) of the studied genes was calculated using the Comparative ∆∆Ct Method, where the amount of specific gene expression is adjusted to GAPDH expression and relative to a reference control [16]. To confirm the precision of the amplification and the absence of primer dimers, melting curve analysis was performed (Figure 2).

We included three technical replicates for every gene, randomly selected from each PCR plate, to ensure accuracy and precision by minimizing errors from pipetting and other experimental variability.

### 2.5. Statistical Analysis

All data were compiled and analyzed using SPSS version 21 (SPSS Inc., Chicago, IL, USA). Continuous variables are presented as means (±standard deviation [SD]), and categorical variables are presented using relative frequency distributions and percentages. The Kolmogorov–Smirnov and Shapiro–Wilk tests were used to test the normality of the distribution of numerical variables. Continuous variables were compared using Student’s *t*-test for parametric data or the Mann–Whitney (U) test for non-parametric data, and categorical data were analyzed using the chi-square (X^2^) test; meanwhile, the Fisher Exact test FEX was used instead when more than 20% of expected values were less than 5.

Spearman correlation was used to measure the strength and direction of relationship between two non-parametric quantitative variables; their values always range between −1 (strong negative relationship) and +1 (strong positive relationship) and the test criteria can be classified according to a Likert scale as the following: r = 0, no correlation; 0.00 > r > 0.3, weak correlation; 0.3 ≥ r > 0.7, moderate correlation; 0.7 ≥ r > 1.00, strong correlation; and r = 1, complete correlation.

To confirm the adequacy of the data for regression analysis, the study applied the formula suggested by Khamis and Kepler (2010) [17]: n >= 20 + 5k, where k represents the number of predictors. Since there are seven independent variables (the 7 I’s of TL), the minimum number of required responses was 55 (n = 20 + 5 (7) = 55). Our study was conducted on 64 eyes of 40 patients with DME to study the response after IVR (divided as responders (43 eyes of 27 patients) and non-responders (21 eyes in 13 patients).

Regarding receiver operating characteristic (ROC) curve analysis, the area under the curve (AUC), specificity, sensitivity, positive predictive value (PPV), negative predictive value (NPV), and accuracy were utilized. Univariate and multivariate logistic regression and odds ratios with 95% CI were used to determine the potency of the tested markers as predictors of non-response to intravitreal Ranibizumab injection. Statistical significance was established at *p* ≤ 0.05.

### 2.6. Sample Calculation Size

Sample size calculation was based on the previous literature [18,19,20]. The minimum required sample size was 34 patients to estimate a supposed medium effect size of 0.586, power of 80%, confidence interval of 95% and alpha error of 0.05; that was increased to 40 patients to compensate for a 15% drop-out rate of follow-up in one group for two repeated measures (before and after treatment), and another matched 40 controls, giving a total of 80 participants equally divided into two groups. The sample size was calculated using G*Power 3.1.9.4 software.

## 3. Results

A flowchart of the study population is shown in Figure 3. Of 97 subjects, 17 were excluded from the study (7 subjects declined consent, and 10 subjects did not meet the inclusion criteria); 80 subjects were included in the study, divided into two groups. Group A included 64 eyes of 40 DME patients. Group B consisted of 40 eyes from 40 healthy controls. According to the response after the first IVR injections, Group A patients were then categorized into non-responders (21 eyes of 13 patients) and responders (43 eyes of 27 patients).

### 3.1. Demographics and Clinical Characteristics of the Study Population (Table 1)

In our study, Group A included 18 males [45%] and 22 females [55%], while Group B included 21 males [52.5%] and 19 females [47.5%]. The mean age was 58.95 ±10.38 and 55.60 ±16.26 in Groups A and B, respectively. There was no statistically significant difference between the studied groups in terms of age [*p* = 0.275] and sex [*p* = 0.502]. In Group A, the mean age of onset of DM was [43.45 ± 10.84], and the duration of DM was [15.58 ± 4.49].

### 3.2. Levels of the Studied Biomarkers Before Intravitreal Injection of Ranibizumab (Table 2)

The levels of lymphocyte activation markers, NETosis, angiogenic, astrocytic, innate immunity, and *miR-135a-5p* biomarkers before IVR were significantly higher in patients with DME than in controls (*p* < 0.05).

### 3.3. Spearman Correlation Regarding the Tested Biomarkers’ Expressions in Blood and Aqueous Before and After Treatment Among the Studied Groups (Table 3)

There was a significant correlation between blood and aqueous samples regarding the tested biomarkers, indicating that blood samples may be as good as aqueous samples to represent the levels of the tested biomarkers.

### 3.4. Relating the Response to IVR with the Demographics and Clinical Characteristics of the Patients with DME (Table 4)

There was no significant difference between non-responders and responders to IVR treatment in terms of age, sex, age of DM onset, and duration of DM [*p* > 0.05]. There was a significant difference between non-responders and responders to IVR treatment regarding HbA1C [*p* = 0.052].

### 3.5. The BCVA, OCT Findings, and Macular Ischemia in the Patients with DME (Table 5)

Before IVR injection, the BCVA and CMT did not differ significantly between responders and non-responders (*p* > 0.05). One month after IVR, the BCVA was considerably better, and the CMT was significantly reduced in responders compared to non-responders. The inner segment/outer segment (IS/OS) junction was significantly disrupted in 53.8% of the non-responders. Macular ischemia (defined as enlarged foveal avascular zone (FAZ)) was significantly present in 38.1% of non-responders (Figure 4 and Figure 5).

### 3.6. The Difference Between the Non-Responders and Responders Regarding the Studied Biomarkers Before IVR (Table 6)

The aqueous levels of *ANGPT1*, *ANGPT2*, *Hexokinase 1*, *ELANE*, *MPO*, and *SEMA4D* were significantly higher in the non-responders.

### 3.7. The Difference Between the Non-Responders and Responders Regarding the Studied Biomarkers One Month After IVR (Table 7)

The aqueous levels of *Hexokinase 1*, *SELL CD62L*, *ELANE*, *MPO*, *VEGF A*, *RAGE* and *miR135a-5p* were significantly lower in the responders (4.46 ± 9.83, 5.44 ± 10.92, 3.83 ± 6.15, 19.62 ± 28.81, 4.21 ± 4.78, 15.64 ± 17.47 and 19.17 ± 60.45) than non-responders (17.84 ± 15.52, 21.94 ± 23.73, 24.66 ± 28.86, 30.07 ± 32.86, 26.82 ± 26.59, 31.65 ± 24.60 and 111.89 ± 156.89), respectively.

### 3.8. Univariate and Multivariate Regression Analysis of Different Parameters Affecting Response to IVR (Table 8a–e)

Univariate regression analysis revealed that the disrupted IS/OS segment, macular ischemia, and aqueous levels of *hexokinase 1*, *SELL CD62L*, *ELANE*, *PAD4*, *MPO*, *VEGF-A*, *SEMA4D*, *GDNF*, *RAGE*, and *miR-135a-5p* after treatment were the most significant factors affecting non-response to IVR (*p* < 0.05). However, multivariate regression analysis revealed that macular ischemia, aqueous levels of *hexokinase 1*, *SELL CD62L*, *ELANE*, *MPO*, *VEGFA*, and *SEMA4D* after treatment were the most significant factors affecting the response to IVR (*p* < 0.05).

### 3.9. The ROC Curve and Area Under the Curve (AUC) to Determine the Cut-Off Point for Each Parameter Affecting the Response to IVR in the Diabetic Patients with DME (Table 9)

The AUCs of CMT reduction after injection were statistically significant using the cut-off point for each marker. At the cut-off point of 10.011%, CMT reduction after injection had 81.5% sensitivity, 100% specificity, and 87.5% accuracy for predicting the response to IVR (Figure 6).

AUCs of *hexokinase 1*, *RCVRN 1*, and *SELL CD62L* aqueous levels were statistically significant using the cut-off point for each marker. Out of these biomarkers that represent the autoimmune element of DME pathophysiology, the best sensitivity (true positives) was 92.3% with ***hexokinase 1***, and the best specificity (true negatives) was 88.9% with ***SELL CD62L*** (Figure 7).

The AUCs of *ELANE*, *PAD4*, and *MPO* aqueous levels were statistically significant using the cut-off point for each marker. Out of these biomarkers that represent the inflammatory element of DME pathophysiology, the highest sensitivity was achieved with ***MPO***, at 100%, and the highest specificity was observed with ***ELANE*,** at 92.6% (Figure 8).

All AUCs of the angiogenic biomarkers were statistically significant using the cut-off point for each marker. Out of these biomarkers that represent the vascular element of DME pathophysiology, the highest sensitivity was 92.3% with ***SEMA4D*,** and the highest specificity was 85.2% with ***SEMA4D* and *VEGF-A*** (Figure 9)

Only the AUCs of *GNDF*, *RAGE*, and *miR-135a-5p* aqueous levels were statistically significant using the cut-off point for each marker. Out of these astrocytic, innate immunity, and inflammasome biomarkers that represent combined vascular and inflammatory elements of DME pathogenesis, the best sensitivity was 84.6% with ***RAGE***, and the best specificity was 74.1% with ***miR-135a-5p*** (Figure 10).

### 3.10. Combined Panels of the Significant Biomarkers in the Aqueous Humor and the Blood Between Responders and Non-Responders (Table 10, Figure 11)

Table 10 and Figure 11 show the differences between the combined panel of the significant six biomarkers (*hexokinase 1*, *SELL CD62L*, *ELANE*, *MPO*, *VEGFA*, and *SEMA4D*) and the short panel of the most sensitive three biomarkers (*hexokinase 1*, *MPO* and *SEMA4D*) in order to reach a short and more condensed panel representing the different components of the DME’s pathology.

## 4. Discussion

Diabetic macular edema is a complex disease driven by various interrelated factors that ultimately compromise the blood–retinal barrier (BRB). This disruption results in fluid accumulation within the retinal layers of the macula, which occurs due to alterations in the structure and function of tight junctions, loss of pericytes, endothelial cell damage, and increased permeability across retinal vessel walls and retinal pigment epithelium cells. When BRB integrity is compromised, fluid begins to leak into the neurosensory retina, accumulating as the inflow exceeds the retina’s drainage capacity [21].

The pathophysiological basis of DME involves persistent hyperglycemia, which promotes the formation of free radicals and advanced glycation end-products (AGEs), leading to oxidative stress and activation of protein kinase C (PKC). This cascade upregulates VEGF-A, a significant factor that enhances vascular permeability and contributes to the development of DME. Additionally, ischemia within the retina, coupled with the inflammatory response, further promotes VEGF-A production. Elevated levels of inflammatory cytokines exacerbate this condition by intensifying vascular permeability and fluid retention within the retinal tissue. Hyperglycemia can also directly elevate PKC and angiotensin II levels, contributing to vasoconstriction and exacerbating hypoxia through endothelin activity [22].

Through these interconnected mechanisms, the management of DME involves various treatment approaches, each targeting specific aspects of the disease’s multifactorial pathophysiology. Key management strategies include anti-VEGF therapies, which are considered the primary treatment option for DME due to their effectiveness in inhibiting VEGF, a crucial factor contributing to the development of DME and proliferative diabetic retinopathy [23].

There are several intravitreal anti-VEGF agents approved by randomized controlled trials (RCT) for treatment of DME: aflibercept (approved by the VIVID and VISTA trials) [23,24,25]; bevacizumab, widely used globally for treating DME due to its lower cost but it is not FDA approved and off label [26,27]; faricimab (approved by YOSEMITE and the RHINE trials) [28,29]; and brolucizumab (approved by KESTREL and KITE trials [30,31].

Ranibizumab is the fragment antigen-binding (Fab) portion of a humanized monoclonal antibody targeting VEGF, specifically inhibiting *VEGF-A*. As only the Fab portion of the full-length immunoglobulin G (IgG) antibody (molecular weight approximately 150 kDa), Ranibizumab has a significantly lower molecular weight of around 48 kDa. This smaller size enhances its diffusion capacity, enabling it to penetrate retinal tissues more efficiently than full-length IgG, which is typically restricted by barriers such as the internal limiting membrane (ILM) and the outer plexiform layer [32].

The clinical effectiveness of Ranibizumab in DME was demonstrated in the RISE and RIDE phase III trials, which were double-masked, sham-controlled, randomized studies. At 24 months, the mean best-corrected visual acuity improvements in RISE were 12.5 letters in the 0.3 mg group and 11.9 letters in the 0.5 mg group, and in RIDE, gains were 10.9 letters in the 0.3 mg group and 12.0 letters in the 0.5 mg group [32,33].

Another line of treating DME is the use of intravitreal steroids, which offer similar efficacy to anti-VEGF therapies [34,35]. However, they are not suitable as first-line options due to their association with risks such as increased intraocular pressure (IOP) and cataract formation [36,37].

Our study aimed to justify the current treatment plan for DME by addressing two key cornerstones. The first objective was to define the response to intravitreal anti-VEGF injection earlier, after the first intravitreal injection, rather than the current plan of assessing the response after 3–5 monthly loading doses. Early identification is crucial for facilitating timely transitions to alternative anti-VEGF therapies or other treatment options, ultimately aimed at reducing the treatment burden and associated costs for patients, particularly in developing countries, and also to save the macula as soon as possible. The responses to IVR are interpreted in the form of BCVA improvement and CMT reduction. The second objective was to search for a suitable clinical–biochemical panel that represents the possible cause/s of non-response in each non-responder, and, thus, change the treatment plan accordingly. The ischemic element (macular ischemia) of the DME process can be easily diagnosed by FFA or optical coherence tomography angiography (OCTA). The difficulty lies in how we can reach the main DME’s pathology in each diabetic patient (as in most patients, the pathology is multifactorial and complex). We attempted to assess several biomarkers that could represent the claimed elements of DME pathophysiology, either in blood or aqueous humor, and use this information to determine the treatment plan accordingly.

We used the receiver operating characteristics (ROC) curve and area under the curve (AUC) to determine the cut-off point for each parameter affecting the response to IVR in diabetic patients with DME.

This study demonstrated that a reduction in CMT of more than 10% from baseline one month after IVR is a cut-off point for categorizing our patients into responders, with 81.5% sensitivity, 100% specificity, and 87.5% accuracy. Turski et al. [18] demonstrated that eyes were classified as responders if CMT reduction was greater than 10% at 4–6 weeks after a single dose of intravitreal bevacizumab. Based on ROC curve analysis, Shah et al. [19] stated that a reduction in CMT by >15% at one month and a CMT reduction of >25% at three months identified eyes that responded to anti-VEGF treatment (sensitivity 75%, specificity 92%). Santos et al. [38] demonstrated that a mean visual improvement of 4.78 and 5.52 ETDRS letters, and a CMT decrease of ≥20% were found after 3 and 6 months of IVR injection. Also, they showed a strong correlation between visual improvement and the degree of CMT reduction.

In our series, the IS/OS junction was disrupted in 52.4% of the eyes of the non-responders. Sharef et al. [39] demonstrated that the integrity of the external limiting membrane and IS/OS of the photoreceptors correlated with BCVA, while also showing a weak and inverse correlation with CMT. Patients with intact photoreceptors at baseline scored a higher final VA than those with disrupted photoreceptor layers. Sujoy et al. [40] found that a broken IS-OS junction on OCT manifests as poor visual acuity, which could indicate an ischemic macula, consistent with a broken FAZ.

In our study, multivariate regression analysis revealed that macular ischemia and aqueous levels of *hexokinase 1*, *SELL CD62L*, *ELANE*, *MPO*, *VEGFA*, and *SEMA4D* were the most significant factors influencing the response to IVR (*p* < 0.05). These results align with our results from the ROC curve and AUC to determine the cut-off point for each parameter affecting the response to IVR in diabetic patients with DME, which concluded that the same biomarkers had the best sensitivity and specificity. Additionally, our series demonstrated a significant correlation between blood and aqueous samples regarding the same biomarkers, suggesting that blood samples might replace aqueous samples as representatives of the biomarker levels. Therefore, we can use these specific biomarkers to provide a clue about the primary pathological component of DME in each patient.

In line with our results regarding macular ischemia as a strong clinical predictor of poor or non-response after IVR injection for DME, Chouhan et al. [20] and Michael et al. [41] identified the optical coherence tomography angiography (OCTA) in patients who had received at least one intravitreal anti-VEGF injection for DME. They found that the responders had a higher vessel density and higher perfusion density in the outer ring, in addition to a full ring at levels of the superficial capillary plexus (SCP). They also observed a lower vessel diameter index in the deep capillary plexus (DCP) in responders when compared to non-responders.

Otani and Kishi [42] stated that *hexokinase 1* is highly expressed in the retinal outer plexiform layer (OPL). Pathological disruption in the OPL in DR patients may lead to the escape of this intracellular protein into the extracellular spaces, along with the breakdown of blood–retinal barriers. This breakdown may allow antigen-presenting cells, such as monocytes or macrophages, to enter the systemic circulation, leading to the subsequent formation of anti-hexokinase1 antibodies [43]. Yoshitake et al. [11] found high levels of anti-*hexokinase 1* IgG in diabetic patients with center-involved DME, suggesting its diagnostic significance.

Regarding *SELL CD62L*, Siddiqui et al. [44] found that levels of *P-selectin* and *L-selectin* were related to diabetic nephropathy, while retinopathy was associated with *L-selectin* only. Additionally, it is noted that cell adhesion molecules and selectins are indicators of microvascular complications among diabetic patients. Circulating levels of cell adhesion molecules (VCAM-1) and selectins (*P-selectin* and *L-selectin*) are associated with the future development of microvascular complications, including diabetic neuropathy, retinopathy, and nephropathy.

The *S100A8* and *S100A9* have inflammatory roles. They are expressed at high levels in neutrophils and low levels in monocytes [45,46,47].

This study demonstrated that *VEGF-A* and *SEMA4D* levels before treatment had a good predictive value for identifying responders to IVR treatment. We found that responders had a higher baseline level of *VEGF A* in their aqueous humor before treatment than non-responders. The non-responders had a significantly higher level of *SEMA4D* in their aqueous than the responders. Concerning this, Tetikoğlu et al. [48] found that the *VEGF-A* mediates angiogenesis by increasing vascular permeability. VEGF expression is significantly elevated in the aqueous humor and vitreous humor of patients with DR and DME, playing a crucial role in the development and progression of these conditions. Shimura et al. [49] and Udaondo et al. [50] found that there were higher baseline aqueous humor concentrations of *VEGF A* in good responders to Ranibizumab therapy compared to non-responders.

Wu J et al. [51] stated that *SEMA4D*, a membrane-bound protein, could be shed from the retinal cells by proteolysis to yield a soluble form (*sSEMA4D*) during different stages of DR. They found that *sSEMA4D* levels were increased in the aqueous fluid of DR patients compared to controls. Also, they stated that the success of anti-VEGF therapy had a negative correlation with levels of *SEMA4D* in the aqueous fluid of DR patients during clinical follow-up. They also suggested that *SEMA4D* levels in aqueous fluid may aid in the early identification of non-responders or those with poor response.

Our study revealed that *MPO* levels were significantly higher in diabetic patients compared to controls. Sinha et al. [52] noted a decrease in visual acuity, which was significantly associated with an increased severity of diabetic retinopathy and higher serum anti-*MPO* antibodies in diabetic patients compared to controls. The increase in anti-*MPO* antibodies is associated with increased expression of *ICAM1* and reactive oxygen species formation, thereby increasing oxidative stress. Accardo et al. [53] also showed that levels of soluble ICAM-1 were found to be higher in anti-MPO antibody-positive than in anti-MPO antibody-negative patients.

In this series, NETosis biomarkers were significantly higher in diabetic patients than in controls. Shurtz-Swirski et al. [54] reported that patients with DR are exposed to increased oxidative stress induced by neutrophils. Park et al. [55] demonstrated that NETs have been formed in DR patients. Circulating NETs, circulating DNA-histone complex, and polymorphonuclear NE can be used as biomarkers reflecting the risk of DR [56,57].

Glial cell line-derived neurotrophic factor (*GDNF*) and neurturin facilitate glutamate uptake by retinal cells, which may help mitigate apoptosis and reflect a compensatory mechanism in response to neuroinflammation and neuronal damage typically seen in diabetic retinopathy [58]. Nishikiori et al. [59] found that the *GDNF* level in the vitreous increases in PDR and is associated with the progression of PDR. Lu et al. [60] found that *RAGE* (Receptor for Advanced Glycation End-products) plays an essential role in the pathogenesis of DR. *RAGE* binding to multiple ligands generates a series of cascade reactions by activating downstream signalling pathways, leading to NF-κB activation, gene transcription of inflammatory mediators, neurodegeneration, and angiogenesis, which are critical in the pathogenesis of DR. Additionally, Saleh et al. [61] reported that accumulation of AGEs in chronic diabetes causes death of pericytes and thickening of the basement membrane, which causes incompetence of vascular walls. AGEs also stimulate *RAGE* expression, leading to the activation of VEGF, which in turn triggers new vessel formation in the retina. Also, AGE induces apoptosis of pericytes through interaction with *RAGE* and *VEGF*.

MicroRNAs are small RNAs, formed from about 22 nucleotides, which do not encode proteins but regulate the expression of other genes. In the search for new biomarkers and therapeutic targets for diabetic retinopathy (DR), miRNA research has focused mainly on cell lines and animal models, with limited application in human samples. Only a small number of studies have assessed miRNA levels in human plasma or serum, predominantly through case-controlled studies involving either type I or type II diabetes [62,63]. There are a few studies discussing the biomarker roles of miRNA in DR and DME [64,65]. Cho et al. [63] reported that several serum miRNAs have been associated with angiogenesis, inflammation, and other conditions that play an essential role in DR pathophysiology; therefore, they may represent biomarkers for DR and DME. Li Y et al. [66] reported that high expression of miR-135a-5p promotes angiogenesis in endothelial cells.

Our study showed that *hexokinase 1*, *MPO*, and *SEMA4D* gave the best sensitivity as predictors of the response to IVR injection; also, the levels of the same biomarkers were significantly higher in the non-responders.

PCR array is a gene expression tool that uses the sensitivity of quantitative PCR (qPCR) in a multi-gene format to analyze the expression levels of a focused panel of pathway- or disease-relevant genes, providing quantitative, pathway-focused data on several genes simultaneously. PCR arrays are more reliable and accurate than other gene expression methods for analyzing biological pathways and disease states [67].

For future directions, we recommend the presence of a PCR array containing *hexokinase 1*, *MPO*, and *SEMA4D* (together with the clinical element of macular ischemia), as this panel will be of high sensitive diagnostic and prognostic value for DME management.

Finally, to our knowledge, there is now a definite etiology-based plan to treat our patients with DME. The current plan is an unlimited number of intravitreal anti-VEGF injections. After that, we may or may not shift to intravitreal steroids without relating this treatment plan to the possible specific patients’ DME pathology. We hope our study will change this concept by

-Starting with a single dose of anti-VEGF, whatever the type, then assessing the response after one month.-If there is a good response, continue injecting the same anti-VEGF agent.-If there is poor or no response, ask the patient for our proposed clinical–biochemical panel, which is composed of FFA or OCTA (to exclude macular ischemia) + PCR array containing the most sensitive genes of *hexokinase 1* (autoimmune), *SEMA4D* (vascular), and *MPO* (inflammatory). According to the results of this panel, we can adjust the treatment accordingly.

There were some limitations in our study. First, the relatively small sample size; however, it may serve as a helpful starting point for studies with larger sample sizes to achieve a more evidence-based treatment for our DME patients. Second, the availability and the high cost of our proposed clinical–biochemical panel to reach an etiology-based treatment of DME; we recommended its use in the non-responders only, at least currently, until this panel is commercially available at a reasonable price such that it could be utilized in routine diagnosis for all DME patients, either responders or non-responders. Third, a group of diabetic patients without DME was not included in the study, in addition to the two studied groups. However, the number of available kits would have been insufficient for testing three groups of patients. Fourth, we depended on FFA in the diagnosis of macular ischemia instead of the newer and more safe technology of OCTA.

## Figures and Tables

**Figure 1 biomedicines-13-02438-f001:**
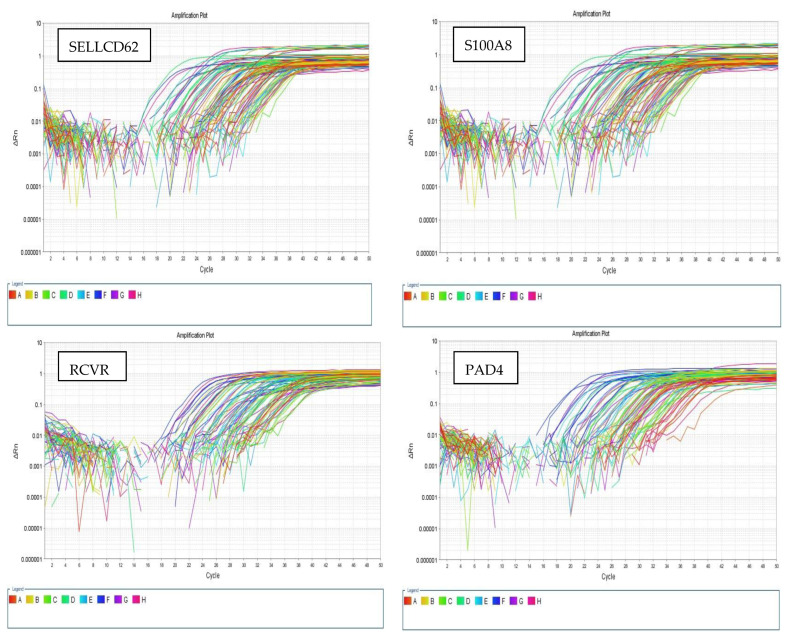
The amplification plots of the tested biomarkers.

**Figure 2 biomedicines-13-02438-f002:**
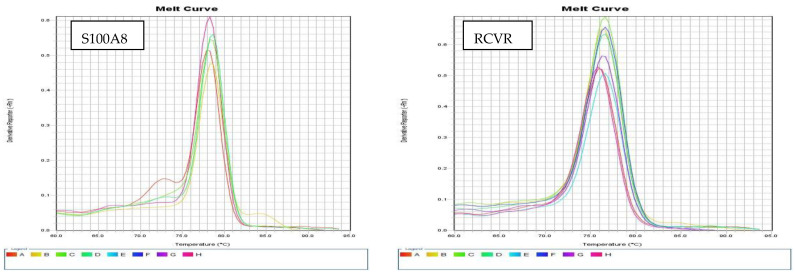
The melting curves of the tested biomarkers.

**Figure 3 biomedicines-13-02438-f003:**
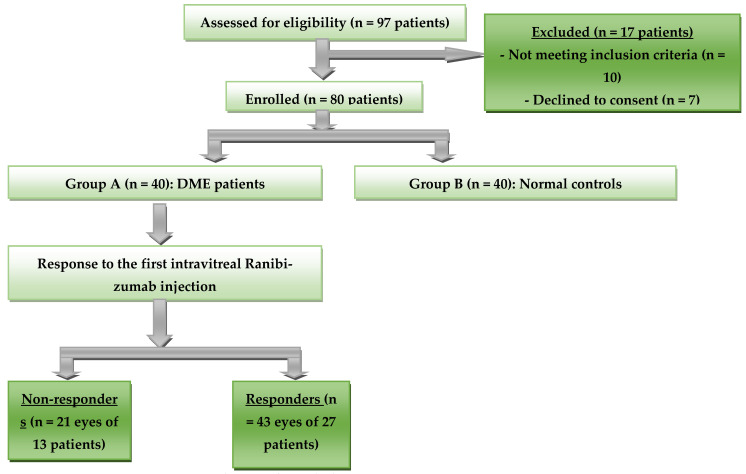
The study design.

**Figure 4 biomedicines-13-02438-f004:**
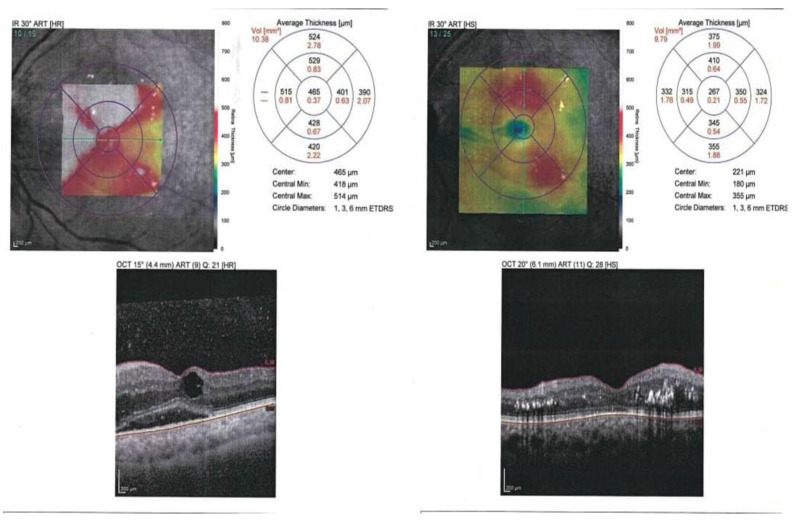
OCT (before and after one month of IVR) of a responder case.

**Figure 5 biomedicines-13-02438-f005:**
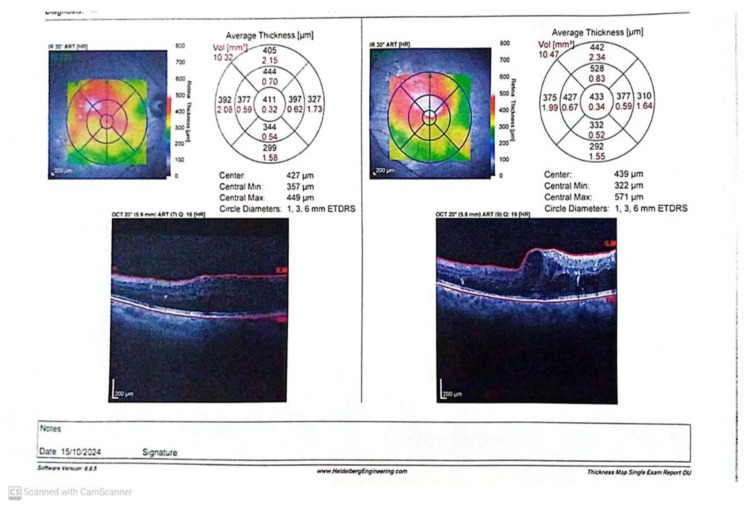
OCT (before and after one month of IVR) of a non-responder case.

**Figure 6 biomedicines-13-02438-f006:**
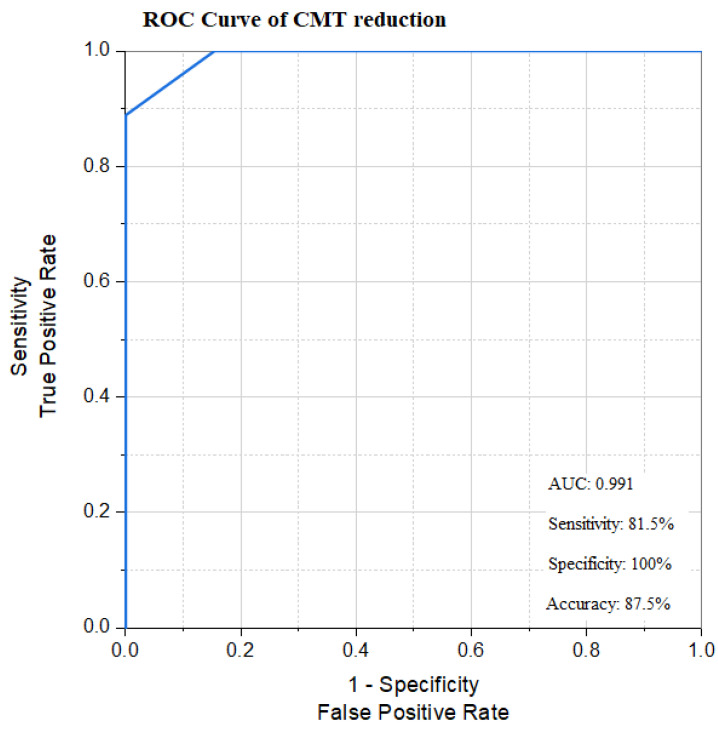
ROC curves of CMT reduction.

**Figure 7 biomedicines-13-02438-f007:**
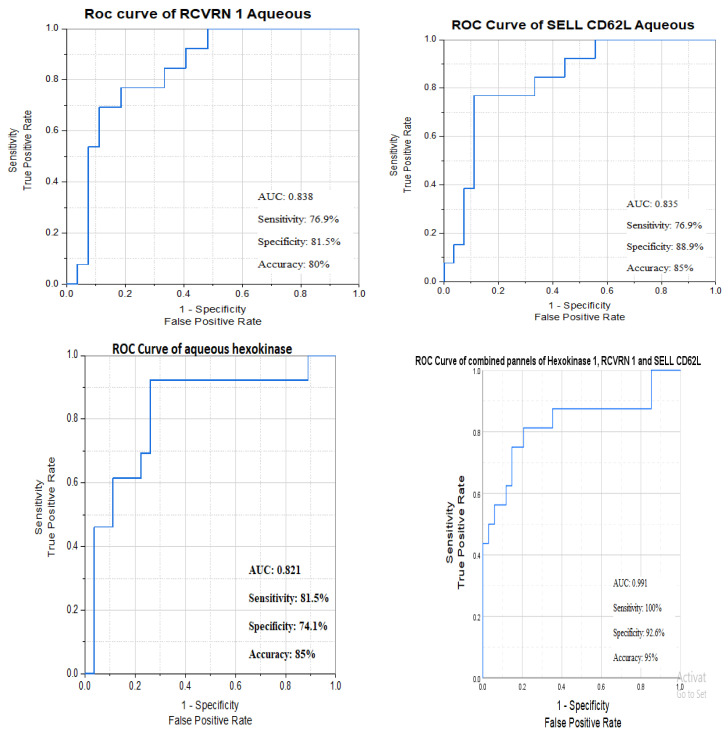
ROC curves of autoimmune biomarkers’ aqueous levels between responders and non-responders.

**Figure 8 biomedicines-13-02438-f008:**
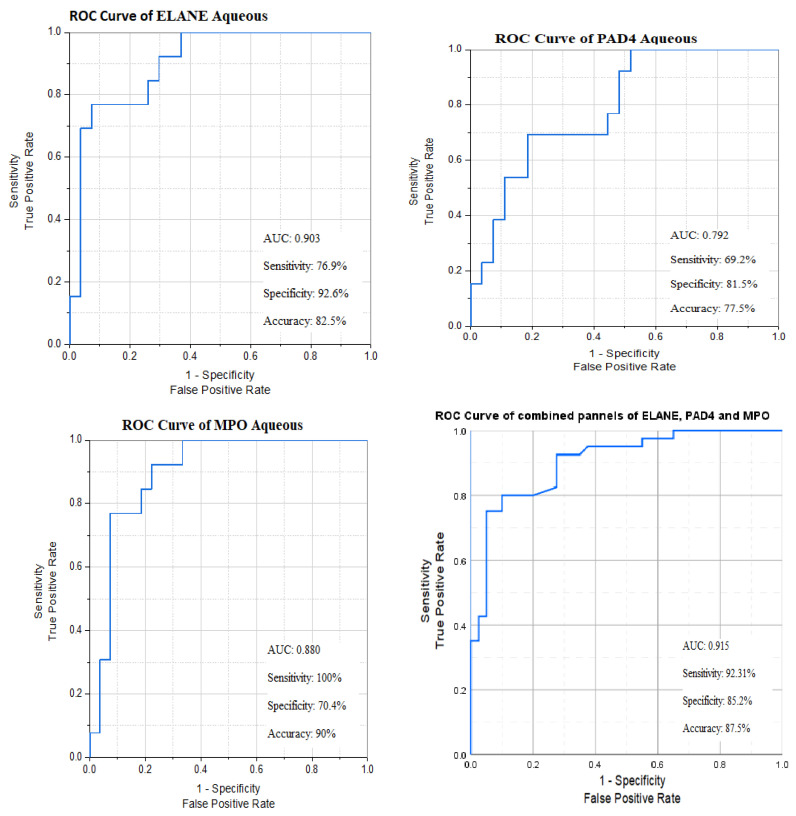
ROC curves of NETosis biomarkers’ aqueous levels between responders and non-responders.

**Figure 9 biomedicines-13-02438-f009:**
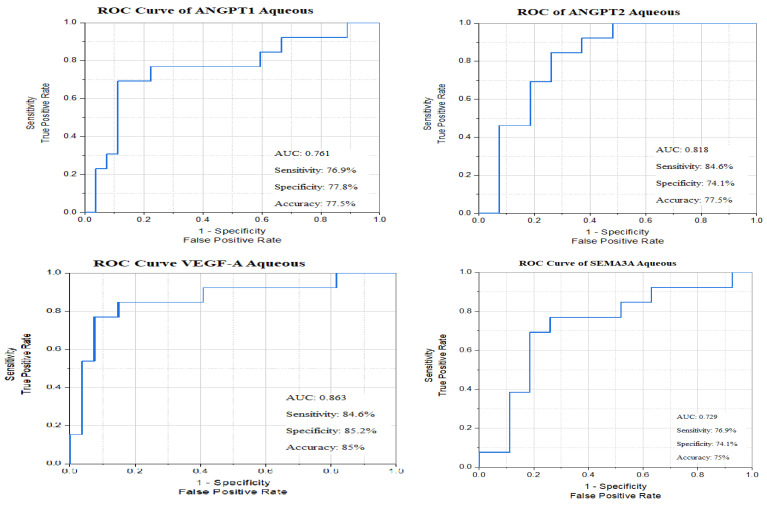
ROC curves of angiogenic biomarkers’ aqueous levels between responders and non-responders.

**Figure 10 biomedicines-13-02438-f010:**
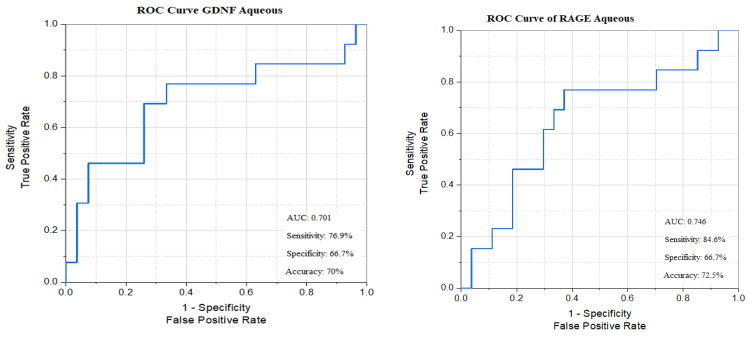
ROC curves of astrocyte, innate immunity, and miR-135a-5p biomarkers’ aqueous levels between responders and non-responders.

**Figure 11 biomedicines-13-02438-f011:**
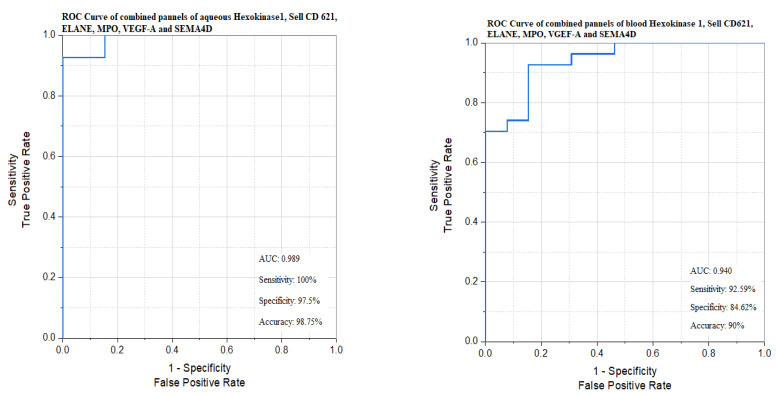
Combined panels of the significant biomarkers in the aqueous humor and blood between responders and non-responders.

**Table 1 biomedicines-13-02438-t001:** Demographics and clinical characteristics of the study populations.

Variable	Group A (DME Patients) (n = 40)		Group B (Controls) (n = 40)		t	*p* Value
	N	%	N	%		
**Age (years)**					1.098	0.275
Mean ± SD	58.95 ± 10.38		55.60 ± 16.26			
**Sex**					0.450	0.502
Male	18	45.0	21	52.5		
Female	22	55.0	19	47.5		
**Age of onset of DM (years)**			--		--	--
Mean± SD	43.45 ± 10.84					
**Duration of DM (years)**			--		--	--
Mean± SD	15.58 ± 4.49					
**HbA1C (%)**			5.13 ± 0.33		21.093	<0.001 *
Mean± SD	7.96 ± 0.78					

**DM:** diabetes mellitus, **BCVA:** best corrected visual acuity, IVR: intravitreal Ranibizumab, **HbA1C:** hemoglobin A1c test, t: independent *t* test, X^2^: chi-square test, *****: significant.

**Table 2 biomedicines-13-02438-t002:** Levels of the studied biomarkers before intravitreal injection of Ranibizumab.

Biomarker	DME patients (Group A) (n = 40)	Controls (Group B) (n = 40)	Mann–Whitney Test	*p* Value
	Mean ± SD	Mean ± SD		
**Lymphocyte activation markers**				
*Hexokinase 1* Blood	13.13 ± 20.08 5.87(2.31–40.99)	0.99 ± 0.12 1.0(0.97–1.26)	0.000	<0.001 *
*Hexokinase 1* Aqueous	9.61 ± 6.65 7.41(1.54–23.96)	0.901 ± 0.17 0.92(0.89–1.25)	0.000	<0.001 *
*SELL CD62L* Blood	17.09 ± 35.53 6.43(1.19–187.66)	0.974 ± 0.19 0.93(0.87–1.35)	9.00	<0.001 *
*SELL CD62L* Aqueous	10.67 ± 19.80 19.8(1.05–97.6)	1.02 ± 0.28 1.195(0.88–0.42)	79.00	<0.001 *
*RCVRN 1* Blood	7.54 ± 7.09 4.36(0.92–22.07)	1.05 ± 0.62 0.95(0.86–2.47	251.50	<0.001 *
*RCVRN 1* Aqueous	10.11 ± 12.23 5.45(1.29–31.74)	1.11 ± 0.22 1.07(0.82–0.76)	6.00	<0.001 *
*S100A8* Blood	9.87 ± 18.70 4.14(1.25–94.43)	0.967 ± 0.10 0.95(0.85–1.24)	0.000	<0.001 *
*S100A8* Aqueous	9.33 ± 17.16 4.89(0.99–90.46)	1.74 ± 0.26 1.75(1.2–2.06)	264.5	<0.001 *
**NETosis biomarkers**				
*ELANE* Blood	13.20 ± 17.12 7.38(0.87–90.07)	0.839 ± 0.22 0.88(0.81–1.3)	18.00	<0.001 *
*ELANE* Aqueous	17.24 ± 28.76 4.28(0.83–100.49)	0.92 ± 0.17 0.87(0.82–1.06)	20.00	<0.001 *
*PAD4* Blood	10.33 ± 17.98 5.01(0.91–70.84)	0.86 ± 0.21 0.88(0.83–1.09)	102.00	<0.001 *
*PAD4* Aqueous	4.74 ± 8.41 2.43(0.02–21.0)	0.83 ± 0.22 0.87(0.79–1.18)	149.00	<0.001 *
*MPO* Blood	19.07 ± 29.99 5.23(0.97–90.29)	0.93 ± 0.14 0.9(0.89–1.09)	40.00	<0.001 *
*MPO* Aqueous	17.34 ± 27.84 5.33(0.81–92.58)	0.80 ± 0.13 0.81(0.84–0.96)	34.00	<0.001 *
**Angiogenic biomarkers**				
*ANGPT1* Blood	16.43 ± 18.57 8.32(0.90–64.95)	0.94 ± 0.20 0.99(0.85–1.17)	160.00	<0.001 *
*ANGPT1* Aqueous	17.93 ± 30.21 10.06(0.9–164.37)	1.75 ± 0.31 1.79(1.06–2.19)	200.00	0.001 *
*ANGPT2* Blood	17.57 ± 20.52 6.79(0.8–80.59)	0.96 ± 0.14 0.99(0.94–1.15)	120.00	0.001 *
*ANGPT2* Aqueous	18.03 ± 25.55 6.63(0.8–93.15)	1.33 ± 0.27 1.35(1.0–1.69)	160.00	<0.001 *
*VEGFA* Blood	25.46 ± 17.70 21.73(5.17–73.46)	1.12 ± 0.27 0.96(0.94–1.65)	0.000	<0.001 *
*VEGF A* Aqueous	35.29 ± 20.93 26.84(10.39–88.22)	1.24 ± 0.42 0.98(0.85–1.96)	0.000	<0.001 *
*SEMA3A* Blood	13.13 ± 17.32 6.13(0.2–61.62)	0.92 ± 0.18 0.94(0.86–1.16)	144.00	<0.001 *
*SEMA3A* Aqueous	14.53 ± 11.82 10.73(0.57–50.57)	0.97 ± 0.13 0.89(0.87–1.17)	39.50	<0.001 *
*SEMA4D* Blood	14.22 ± 21.90 6.82(0.18–42.4)	1.45 ± 0.66 1.43(0.8–2.42)	116.00	<0.001 *
*SEMA4D* Aqueous	15.22 ± 17.59 5.76(1.65–63.84)	1.35 ± 0.69 1.47(0.89–2.64)	36.00	<0.001 *
**Astrocytic Factor**				
*GDNF* Blood	1.15 ± 0.62 1.22(0.99–1.85)	9.83 ± 12.62 6.29 (0.84–73.69)	36.00	<0.001 *
*GDNF* Aqueous	1.23 ± 0.52 1.03(0.78–1.99)	21.27 ± 17.06 14.87(0.93–78.33)	97.00	<0.001 *
**Innate Immunity and Inflammasomes**				
*TXNIP* Blood	14.03 ± 16.27 11.53(0.83–79.99)	1.05 ± 0.47 1.05(0.89–1.97)	176.00	<0.001 *
*TXNIP* Aqueous	16.63 ± 22.45 6.26(0.79–114.95)	1.39 ± 0.47 1.39(0.98–1.98)	103.00	<0.001 *
*NLRP3* Blood	14.44 ± 22.92 5.09(0.33–66.0)	1.12 ± 0.51 0.96(0.93–2.14)	172.00	<0.001 *
*NLRP3* Aqueous	13.22 ± 18.86 3.91(0.81–82.2)	1.30 ± 0.74 1.29(0.82–2.87)	368.50	<0.001 *
*RAGE* Blood	13.37 ± 18.94 5.71(1.09–60.87)	1.48 ± 0.75 1.45(0.82–2.75)	111.50	<0.001 *
*RAGE* Aqueous	22.09 ± 50.27 6.96(0.91–302.29)	0.95 ± 0.27 0.95(0.81–1.38)	52.00	<0.001 *
*miR-135a-5p* Blood	11.45 ± 17.59 3.51(0.80–60.12)	1.85 ± 1.05 1.66(0.80–5.33)	507.50	0.005 *
*miR-135a-5p* Aqueous	57.66 ± 144.20 15.27(0.87–903.07)	7.78 ± 4.51 7.69(0.85–16.81)	549.00	0.016 *

t: independent test, * significant.

**Table 3 biomedicines-13-02438-t003:** Spearman correlation regarding the tested biomarkers’ expressions in blood and aqueous before and after treatment among the studied groups.

	Before Treatment		After Treatment	
	R	*p* Value	R	*p* Value
**Lymphocyte Activation Markers**				
*Hexokinase 1*	0.849	<0.0001 **	0.621	<0.0001 **
*RCVRN 1*	0.581	<0.0001 **	0.191	0.237
*SELL CD62L*	0.664	<0.0001 **	0.403	0.010 *
*S100A8*	0.646	<0.0001 **	0.157	0.334
**NETosis Biomarkers**				
*ELANE*	0.739	<0.0001 **	0.654	<0.0001 **
*PAD4*	0.668	<0.0001 **	0.446	0.004 **
*MPO*	0.698	<0.0001 **	0.685	<0.0001 **
**Angiogenic Biomarkers**				
*ANGPT1*	0.734	<0.0001 **	0.597	<0.0001 **
*ANGPT2*	0.705	<0.0001 **	0.622	<0.0001 **
*VEGF A*	0.969	<0.0001 **	0.624	<0.0001 **
*SEMA3A*	0.691	<0.0001 **	0.340	0.032 *
*SEMA4D*	0.747	<0.0001 **	0.365	0.021 *
**Astrocytic Factors**				
*GDNF*	0.913	<0.0001 **	0.467	0.002 **
**Innate Immunity and Inflammasomes**				
*TXNIP*	0.641	<0.0001 **	0.036	0.825
*NLRP3*	0.559	<0.0001 **	0.007	0.967
*RAGE*	0.599	<0.0001 **	0.276	0.085
*miR-135a-5p*	0.316	0.004 **	−0.062	0.706

R: Spearman correlation coefficient, *: significant at *p* < 0.05, **: highly significant at *p* < 0.01.

**Table 4 biomedicines-13-02438-t004:** Relating the response to IVR with the demographics and clinical characteristics of the patients with DME.

Variable	Non-Responders (n = 21 Eyes of 13 Patients)		Responders (n = 43 Eyes of 27 Patients)		t	*p* Value
	N	%	N	%		
**Age (years)**					0.594	0.556
Mean± SD	60.15 ± 7.16		58.37 ± 11.71			
**Sex**					0.333	0.564
Male	5	38.5	13	48.1		
Female	8	61.5	14	51.9		
**Age of onset of DM (years)**					1.065	0.294
Mean± SD	45.69 ± 7.47		42.37 ± 12.11			
**Duration of DM (years)**					0.967	0.341
Mean± SD	14.69 ± 3.50		16.00 ± 4.90			
**HbA1C (%)**					U = 108.0	0.052 *
Mean± SD	8.36 ± 0.95		7.76 ± 0.61			

**DM:** diabetes mellitus, **HbA1C:** hemoglobin A1c test, t: independent *t* test, X^2^: chi-square test, **U:** Mann–Whitney test. * Significant.

**Table 5 biomedicines-13-02438-t005:** The BCVA, OCT findings, and macular ischemia in the patients with DME.

Variable	Non-Responders (n = 21 Eyes)		Responders (n = 43 Eyes)		t	*p* Value
	N	%	N	%		
**BCVA (Log MAR) before IVR**					0.350	0.729
Mean± SD	0.88 ± 0.11		0.87 ± 0.13			
**BCVA (Log MAR) after IVR**					7.838	<0.001 *
Mean± SD	0.86 ± 0.25		0.55 ± 0.15			
**CMT (microns)**					0.637	0.529
**before IVR**						
Mean± SD	536.00 ± 84.52		516.44 ± 102.93			
**CMT (microns)**					4.702	<0.001 *
**1 month after IVR**						
Mean± SD	514.92 ± 89.55		378.78 ± 77.35			
**IS/OS segment**					X^2^ = 27.692	<0.001 *
Intact	10	47.6	43	100.0		
Disrupted	11	52.4	0	0.0		
**Macular ischemia**					FE = 11.868	0.001 *
No ischemia	13	61.9	43	100.0		
Ischemia	8	38.1	0	0.0		

**BCVA:** best corrected visual acuity, IVR: intravitreal Ranibizumab, **CMT:** central macular thickness, t: independent *t* test, X^2^: chi-square test, * significant.

**Table 6 biomedicines-13-02438-t006:** The difference between the non-responders and responders regarding the studied biomarkers before IVR.

Biomarker	Non-Responders (n = 21 Eyes)	Responders (n = 43 Eyes)	Mann–Whitney Test	*p* Value
	Mean ± SD	Mean ± SD		
**Lymphocyte Activation Markers**				
*Hexokinase 1* Blood	13.41 ± 9.18 13.0(4.11–40.99)	12.99 ± 23.79 12.57(2.31–11.25)	174.00	0.732
*Hexokinase 1* Aqueous	16.58 ± 6.73 16.87(1.98–23.98)	6.25 ± 3.06 5.95(1.54–14.76)	33.00	**<0.001 ***
*RCVRN 1* Blood	9.18 ± 9.26 3.2(1.03–22.07)	6.76 ± 7.44 2.76(0.92–14.2)	147.00	0.424
*RCVRN 1* Aqueous	11.24 ± 9.09 6.88(1.97–31.74)	9.56 ± 13.61 8.15(1.29–21.61)	154.00	0.573
*SELL CD62L* Blood	18.95 ± 50.78 4.18(1.19–187.66)	16.20 ± 26.47 5.45(1.36–38.48)	147.00	0.424
*SELL CD62L* Aqueous	20.86 ± 32.70 5.93(1.1–97.65)	5.76 ± 4.21 5.05(1.05–16.89)	142.50	0.345
*S100A8* Blood	16.40 ± 32.18 4.0(2.17–94.43)	6.73 ± 3.88 5.84(1.25–13.46)	122.00	0.127
*S100A8* Aqueous	17.41 ± 28.61 5.01(0.9–90.46)	5.43 ± 3.94 4.83(0.83–14.64)	146.50	0.407
**NETosis Biomarkers**				
*ELANE* Blood	22.92 ± 27.27 10.22(2.62–90.07)	8.52 ± 5.16 6.99(0.87–26.64)	128.00	0.177
*ELANE* Aqueous	44.25 ± 38.80 5.62(1.49–100.49)	4.24 ± 2.22 3.19(0.83–9.65)	88.00	**0.011 ***
*PAD4* Blood	12.72 ± 19.06 5.01(0.31–70.84)	9.19 ± 17.70 4.96(0.81–17.94)	164.00	0.754
*PAD4* Aqueous	5.46 ± 5.70 2.47(0.95–21.00)	4.39 ± 9.52 2.28(0.82–6.68)	121.00	0.120
*MPO* Blood	21.79 ± 34.22 4.43(1.15–90.29)	17.76 ± 28.34 7.36(0.81–32.36)	147.00	0.424
*MPO* Aqueous	38.38 ± 39.99 4.85(0.86–92.58)	7.21 ± 9.83 3.33(0.81–55.31)	92.50	**0.014 ***
**Angiogenic Biomarkers**				
*ANGPT1* Blood	20.97 ± 26.30 4.01(0.85–64.95)	14.25 ± 13.53 8.82(0.89–54.45)	158.00	0.628
*ANGPT1* Aqueous	15.28 ± 30.74 7.54(0.93–164.37)	23.42 ± 29.50 13.58(0.83–113.55)	119.00	0.106
*ANGPT2* Blood	21.83 ± 24.99 7.99(0.9–76.13)	15.52 ± 18.17 5.59(0.83–80.59)	168.50	0.842
*ANGPT2* Aqueous	14.67 ± 23.58 6.25(0.9–93.15)	25.01 ± 28.95 16.06(1.2–75.87)	160.00	0.669
*VEGFA* Blood	21.31 ± 22.85 21.01(5.17–55.18)	27.45 ± 14.71 24.33(10.33–73.46)	151.00	0.493
*VEGF A* Aqueous	29.63 ± 24.52 28.66(10.39–69.58)	38.01 ± 18.87 32.45(14.74–88.22)	192.00	0.790
*SEMA3A* Blood	10.53 ± 14.79 5.37(0.86–57.95)	18.53 ± 21.30 8.24(0.9–61.82)	145.00	0.391
*SEMA3A* Aqueous	12.09 ± 9.59 18.08(2.0–29.37)	19.59 ± 14.62 18.08(0.87–50.57)	124.50	0.142
*SEMA4D* Blood	17.99 ± 4.60 6.22(0.18–42.4)	12.40 ± 2.71 5.37(0.88–15.99)	150.00	0.475
*SEMA4D* Aqueous	31.66 ± 2.34 15.69(5.4–63.84)	7.31 ± 8.56 3.85(1.65–42.58)	39.00	**0.001 ***
**Astrocytic Factor**				
*GDNF* Blood	8.07 ± 8.19 7.21(0.04–24.03)	10.68 ± 14.34 5.79(1.77–73.69)	150.0	0.475
*GDNF* Aqueous	17.28 ± 14.35 19.05(0.03–42.37)	23.19 ± 18.15 14.46(5.23–78.33)	151.00	0.493
**Innate Immunity and Inflammasomes**				
*TXNIP* Blood	16.41 ± 25.57 2.3(0.03–79.99)	12.89 ± 9.56 12.87(0.79–34.81)	125.00	0.151
*TXNIP* Aqueous	20.23 ± 19.50 12.98(0.29–55.29)	14.89 ± 23.89 5.47(1.04–114.95)	136.50	0.264
*NLRP3* Blood	15.26 ± 18.46 11.18(0.33–56.67)	14.04 ± 25.10 10.6(0.5–66.83)	164.00	0.754
*NLRP3* Aqueous	20.09 ± 25.17 3.45(0.01–82.2)	9.91 ± 14.37 4.07(0.01–55.59)	170.00	0.887
*RAGE* Blood	16.17 ± 20.11 6.16(2.04–58.88)	12.03 ± 18.59 5.66(1.09–60.87)	159.00	0.648
*RAGE* Aqueous	33.53 ± 81.71 11.05(0.05–302.29)	16.58 ± 24.75 6.93(1.03–99.79)	170.00	0.887
*miR-135a-5p* Blood	16.33 ± 25.31 3.15(0.80–60.12)	9.11 ± 12.3 4.82(0.8–60.12)	160.50	0.669
*miR-135a-5p* Aqueous	68.43 ± 171.76 22.04(1.61–903.07)	35.28 ± 53.48 19.02(0.87–147.77)	129.00	0.187

* Significant.

**Table 7 biomedicines-13-02438-t007:** The difference between the non-responders and responders regarding the studied biomarkers one month after IVR.

Biomarker	Non-Responders (n = 21 Eyes)	Responders (n = 43 Eyes)	Mann-Whitney Test	*p* Value
	Mean ± SD	Mean ± SD		
**Lymphocyte Activation Markers**				
*Hexokinase* 1 Aqueous	17.84 ± 15.52 20.46(0.87–43.07)	4.46 ± 9.83 1.0(0.82–44.92)	63.00	0.001 *
*RCVRN 1* Aqueous	11.38 ± 7.18 12.3(1.28–24.76)	9.00 ± 22.15 11.04(0.9–66.76)	160.00	0.616
*SELL CD62L* Aqueous	21.94 ± 23.73 22.01(1.79–93.98)	5.44 ± 10.92 1.94(0.81–50.87)	58.00	0.000 *
*S100A8* Aqueous	19.67 ± 31.79 2.77(0.86–90.57)	5.17 ± 3.32 4.31(0.99–15.65)	151.00	0.493
**NETosis Biomarkers**				
*ELANE* Aqueous	24.66 ± 28.86 13.53(2.63–92.19)	3.83 ± 6.15 1.41(0.81–30.32)	34.00	0.000 *
*PAD4* Aqueous	5.77 ± 4.72 4.21(1.49–15.83)	4.23 ± 9.64 3.58(0.82–10.72)	147.00	0.437
*MPO* Aqueous	30.07 ± 32.86 12.53(3.4–95.7)	19.62 ± 28.81 2.05(0.8–66.7)	42.00	0.000 *
**Angiogenic Biomarkers**				
*ANGPT1* Aqueous	9.82 ± 20.91 13.55(0.9–96.87)	22.50 ± 20.48 15.82(0.88–55.56)	136.00	0.238
*ANGPT2* Aqueous	10.28 ± 22.92 2.76(0.8–89.94)	23.63 ± 25.16 3.51(0.8–89.94)	122.0	0.127
*VEGF A* Aqueous	26.82 ± 26.59 14.7(0.89–56.42)	4.21 ± 4.78 2.19(0.84–21.44)	48.00	0.000 *
*SEMA3A* Aqueous	11.53 ± 22.84 8.93(0.91–90.69)	18.59 ± 29.24 10.69(1.34–112.57	142.00	0.345
*SEMA4D* Aqueous	28.91 ± 23.63 15.73(4.22–37.56)	14.47 ± 23.24 13.48(1.57–16.39)	159.00	0.572
**Astrocytic Factor**				
*GDNF* Aqueous	5.79 ± 9.21 1.44(0.82–39.90)	16.56 ± 15.48 8.63(0.88–45.60)	34.00	0.000 *
**Innate Immunity and Inflammasomes**				
*TXNIP* Aqueous	20.62 ± 15.08 16.39(1.84–49.84)	14.55 ± 13.65 9.36(0.93–53.61)	122.00	0.127
*NLRP3* Aqueous	20.84 ± 27.38 8.51(0.82–83.28)	10.00 ± 20.80 4.25(1.28–96.54)	142.00	0.345
*RAGE* Aqueous	31.65 ± 24.60 21.57(3.80–93.48)	15.64 ± 17.47 10.65(0.99–70.58)	89.0	0.012 *
*miR-135a-5p* Aqueous	111.89 ± 156.89 29.15(0.92–366.1)	19.17 ± 60.45 1.57(0.8–315.58)	103.50	0.036 *

* significant.

**Table 8 biomedicines-13-02438-t008:** (**a**): Univariate and multivariate regression analysis of basic parameters and lymphocyte activation markers affecting response to IVR. (**b**): Univariate and multivariate regression analysis of basic parameters and NETosis biomarkers affecting response to IVR. (**c**): Univariate and multivariate regression analysis of basic parameters and angiogenic biomarkers affecting response to IVR. (**d**): Univariate and multivariate regression analysis of basic parameters and astrocytic factors affecting response to IVR. (**e**): Univariate and multivariate regression analysis of basic parameters and innate immunity and inflammasomes affecting response to IVR.

(a)				
Univariate Logistic Regression			Multivariate Logistic Regression	
Variable	OR (95% C.I.)	*p* Value	OR (95% C.I.)	*p* Value
Age (years)	0.981 (0.911–1.056)	0.608		
Sex	0.762 (0.178–3.262)	0.565		
Duration of DM	1.074 (0.900–1.282)	0.370		
BCVA	0.003 (0.011–617.41)	0.994		
HbA1c (%)	0.316 (0.105–0.955)	0.041 *	0.182 (0.022–1.499)	0.113
Disrupted (IS/OS segment)	11.556 (1.137–117.434)	0.045 *	57.61 (0.780–4254.75)	0.065
CMT	0.992 (0.985–0.999)	0.039 *	0.994 (0.984–1.004)	0.256
Macular ischemia	16.250 (6.648–64.243)	0.001 *	48.55 (1.732–1360.92)	0.022 *
**Lymphocyte Activation Markers**				
*Hexokinase 1* aqueous	0.922 (0.867–0.981)	0.010 *	0.915 (0.834–0.984)	**0.049 ***
*RCVRN 1* aqueous	0.954 (0.890–1.024)	0.194		
*SELL CD62L* aqueous	0.921 (0.860–0.986)	0.018 *	0.963 (0.876–0.988)	**0.044 ***
*S100A8* aqueous	0.947 (0.882–1.016)	0.129		
**(b)**				
**Univariate Logistic Regression**			**Multivariate Logistic Regression**	
**Variable**	**OR (95% C.I.)**	** *p* ** ** Value**	**OR (95% C.I.)**	** *p* ** ** Value**
**HbA1c (%)**	0.316 (0.105–0.955)	0.041 *	0.101 (0.004–2.565)	0.165
Disrupted (IS/OS segment)	11.556 (1.137–117.434)	0.045 *	57.895 (0.031–10,676.32)	0.290
**CMT**	0.992 (0.985–0.999)	0.039 *	0.996 (0.983–1.009)	0.517
Macular ischemia	16.250 (6.648–64.243)	0.001 *	24.93 (0.902–689.24)	0.058
**NETosis Biomarkers**				
*ELANE* aqueous	0.839 (0.736–0.955)	0.008 **	0.863 (0.745–0.984)	**0.027 ***
*PAD4* aqueous	0.723 (0.545–0.960)	0.025 *	0.0.733 (0.408–1.319)	0.600
*MPO* aqueous	0.952 (0.914–0.991)	0.018 *	0.962 (0.885–0.993)	**0.030 ***
**(c)**				
**Univariate Logistic Regression**			**Multivariate Logistic Regression**	
**Variable**	**OR (95% C.I.)**	** *p* ** ** Value**	**OR (95% C.I.)**	** *p* ** ** Value**
HbA1c (%)	0.316 (0.105–0.955)	0.041 *	0.024 (0.00–1.790)	0.090
Disrupted (IS/OS segment)	11.556 (1.137–117.434)	0.045 *	2954.766 (0.976–903,264)	0.051
CMT	0.992 (0.985–0.999)	0.039 *	0.981 (0.959–1.003)	0.084
Macular ischemia	16.250 (6.648–64.243)	0.001 *	206.337 (0.591–72,086.56)	0.074
**Angiogenic Biomarkers**				
*ANGPT1* aqueous	0.973 (0.941–1.06)	0.102		
*ANGPT2* aqueous	0.978 (0.951–1.006)	0.123		
*VEGF A* aqueous	0.970 (0.677–0.922)	0.003 **	0.813 (0.671–0.987)	**0.036 ***
*SEMA3A* aqueous	0.989 (0.964–1.015)	0.409		
*SEMA4D* aqueous	0.776 (0.660–0.913)	0.002 **	0.776 (0.632–0.953)	**0.015 ***
**(d)**				
**Univariate Logistic Regression**			**Multivariate Logistic Regression**	
**Variable**	**OR (95% C.I.)**	** *p* ** ** Value**	**OR (95% C.I.)**	** *p* ** ** Value**
HbA1c (%)	0.316 (0.105–0.955)	0.041 *	0.247 (0.043–1.418)	0.117
Disrupted (IS/OS segment)	11.556 (1.137–117.434)	0.045 *	26.01(0.274–2481.39)	0.160
CMT	0.992 (0.985–0.999)	0.039 *	0.991(0.981–1.001)	0.055
Macular ischemia	16.250 (6.648–64.243)	0.001 *	21.966 (0.938–514.629)	0.055
**Astrocytic Factor**				
*GDNF* aqueous	0.931 (0.877–0.989)	0.021 *	0.956 (0.911–1.003)	0.054
**(e)**				
**Univariate Logistic Regression**			**Multivariate Logistic Regression**	
**Variable**	**OR (95% C.I.)**	** *p* ** ** Value**	**OR (95% C.I.)**	** *p* ** ** Value**
HbA1c (%)	0.316 (0.105–0.955)	0.041 *	0.117 (0.010–1.440)	0.094
Disrupted (IS/OS segment)	11.556 (1.137–117.434)	0.045 *	34.698 (0.532–2263.46)	0.096
CMT	0.992 (0.985–0.999)	0.039 *	0.987 (0.975–1.004)	0.059
Macular ischemia	16.250 (6.648–64.243)	0.001 *	0.724 (0.003–162.422)	0.907
**Innate Immunity and Inflammasomes**				
*TXNIP* aqueous	0.971 (0.927–1.017)	0.212		
*NLRP3* aqueous	0.981 (0.954–1.009)	0.189		
*RAGE* aqueous	0.963 (0.928–0.999)	0.042 *	0.930 (0.859–1.007)	0.099
*miR-135a-5p* aqueous	0.991 (0.983–0.999)	0.035 *	0.985 (0.966–1.005)	0.055

* Significant. ** Highly significant.

**Table 9 biomedicines-13-02438-t009:** The ROC curve and area under the curve (AUC) to determine the cut-off point for each parameter affecting the response to IVR in diabetic patients with DME.

Parameter	AUC	Significance	Best Cut-Off Point	Sensitivity (%)	Specificity (%)	PPV (%)	NPV (%)	Accuracy (%)
CMT Reduction	0.991	<0.0001	10.011%	81.5%	100%	100%	72.22%	87.5%
*Hexokinase 1* Aqueous	0.821	0.001	2.36	92.3%	74.1%	73.33%	92%	85%
*RCVRN 1* Aqueous	0.838	0.001	5.75	76.9%	81.5%	88%	66.67%	80%
*SELL CD62L* Aqueous	0.835	0.001	5.49	76.9%	88.9%	88.9%	76.92%	85%
*S100A8* Aqueous	0.430	0.479						
*ELANE* Aqueous	0.903	<0.0001	9.78	76.9%	92.6%	95.45%	66.67%	82.5%
*PAD4* Aqueous	0.792	0.003	3.785	69.2%	81.5%	84.62%	64.29%	77.5%
*MPO* Aqueous	0.880	<0.0001	3.275	100%	70.4%	87.1%	100%	90%
*ANGPT1* Aqueous	0.761	0.008	9.485	76.9%	77.8%	87.5%	62.5%	77.5%
*ANGPT2* Aqueous	0.818	0.001	1.090	84.6%	74.1%	67.5%	90.91%	77.5%
*VEGF-A* Aqueous	0.863	<0.001	8.25	84.6%	85.2%	92%	73.33%	85%
*SEMA3A* Aqueous	0.729	0.020	8.14	76.9%	74.1%	58.82%	86.96%	75%
*SEMA4D* Aqueous	0.912	<0.001	9.315	92.31%	85.2%	75%	95.83%	87.5%
*GDNF* Aqueous	0.701	0.042	3.195	76.9%	66.7%	52.63%	85.71%	70%
*TXNIP* Aqueous	0.602	0.122						
*NLRP3* Aqueous	0.595	0.333						
*RAGE* Aqueous	0.746	0.012	13.45	84.6%	66.7%	55%	90%	72.5%
*miR-135a-5p* Aqueous	0.746	0.012	7.5335	61.5%	74.1%	53.33%	80%	70%

**Table 10 biomedicines-13-02438-t010:** Combined panels of the significant biomarkers in the aqueous humor and blood between responders and non-responders.

Parameter	AUC	Significance	Best Cut-Off Point	Sensitivity (%)	Specificity (%)	PPV (%)	NPV (%)	Accuracy (%)
Combined 6 aqueous	0.989	<0.0001 *	-------	100%	97.5%	97.56%	100%	98.75%
Combined 6 blood	0.940	<0.0001 *	-------	92.59%	84.62%	92.59%	84.62%	90%
Combined 3 aqueous	0.952	<0.0001 *	-------	92.6%	92.3%	96.15%	85.71%	92.5%
Combined 3 blood	0.900	<0.0001 *	--------	81.48%	76.92%	88%	66.67%	80%

* Significant.

## Data Availability

The datasets created and/or analyzed for the current investigation are not publicly accessible due to the privacy of this pilot study. Still, they are obtainable from the corresponding author upon a valid request.

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
