# Peer review of "Re-Evaluating the Treatment Plan for Diabetic Macular Edema Based on Early Identification of Response and Possible Biochemical Predictors of Non-Response After the First Intravitreal Ranibizumab Injection"

_biomedicines, 2025, doi:10.3390/biomedicines13102438_

Round 1
Reviewer 1 Report
Comments and Suggestions for Authors
This study addresses an important gap in diabetic macular edema management, but requires improvements in clarity, methodological transparency, and presentation to maximize impact and facilitate understanding by clinicians and researchers.
- The figures required significant improvement, which is too poor in quality.
- The manuscript contains several grammatical errors and awkward phrasing (e.g., repeated use of sentence fragments, inconsistent tense). A thorough professional language editing is recommended to improve readability and academic tone.
- Sections such as abstract, introduction, methods, and results should be more clearly delineated with consistent formatting. Adding subheadings within major sections would improve navigation.
- The objective statement should be condensed and clarified at the start of the Introduction. Clearly define study aims and hypotheses, specifying what “individualizing the treatment” entails for diabetic macular edema management.
- More detail is needed on sample size calculation/rationale to justify the choice of 40 patients per group.
- Specify the criteria for categorizing responders and non-responders, including rationale for the chosen cut-offs (e.g., >10% CMT reduction, >5 ETDRS letters improvement).
- Describe randomization procedures, if any, and how bias was minimized during group allocation.
- The methods for biomarker measurement (PCR conditions, sample handling, blinding to response status) should be described more comprehensively.
- Expand on the statistical methods: how multiple comparisons were addressed, and whether the chosen tests were appropriate for all variables.
- For regression analyses, report the number of events per variable to evaluate the robustness of multivariate models.
- Include effect sizes and confidence intervals in tables and text for greater transparency.
- Tables are frequently split and hard to read, with inconsistent use of units, decimals, and statistical notation. Standardize tables and consider including key findings in figures.
- Provide the main findings in a clearer narrative, including the clinical implications (how many patients are affected, magnitude of differences).
- The ROC analyses and cut-off values should be visually summarized in figures for greater clarity.
- The Discussion section is lengthy and at times repetitive. Focus on synthesizing key findings, comparing with relevant published studies, and highlighting novel contributions.
- Clearly address study limitations; including sample size, generalizability, and biomarker availability; as well as future research directions.
- Emphasize how your results may translate into clinical practice or trial design for diabetic macular edema.
- Clearly highlight the novelty; how early identification of non-response and related biochemical predictors can alter treatment paradigms for diabetic macular edema.
- Summarize with a concise statement of the clinical or scientific impact at the end of the Discussion.
Reviewer 2 Report
Comments and Suggestions for Authors
The manuscript raises an important clinical question about redefining DME treatment pathways by assessing response after the first ranibizumab injection rather than waiting for several loading doses. The concept is timely, but several issues need clarification and strengthening:
- Sample size and statistical robustness
The study included only 40 DME patients, subdivided into 27 responders and 13 non-responders (Table 4). Such small subgroups limit the reliability of regression and ROC analyses (Sections 3.6–3.9). Please temper the conclusions and clearly acknowledge the exploratory nature of these findings. - Choice of control group
Group B comprised cataract patients (p. 2, lines 25–26). While matched for age and sex, these were not diabetic controls. This weakens the interpretation of biomarker differences (Table 2), as it is unclear whether elevations reflect DME specifically or diabetes itself. A stronger justification and discussion are needed. - Definition of response
Response was defined as CMT reduction >10% and BCVA gain >5 ETDRS letters after one month (p. 3, lines 117–120). This differs from standard protocols (e.g. DRCR.net, pivotal trials) that assess response after 3–6 months. Please justify why one month was chosen, and consider whether short-term fluctuations might misclassify patients. - Overload of biomarker data
Tables 2, 6, and 7 contain extensive gene expression data, but the discussion does not sufficiently highlight which findings are clinically relevant. For example, ELANE, MPO, VEGF-A, and SEMA4D appear important (Table 8), yet weaker or inconsistent associations receive equal weight. Streamlining results to emphasise the strongest predictors would improve clarity. - Clinical applicability
Many proposed biomarkers require aqueous sampling and advanced molecular assays (p. 4, lines 137–179), which is not practical in routine practice. Please highlight which markers have the best potential for translation into accessible, non-invasive clinical testing. - Imaging findings
The association of macular ischaemia (FAZ enlargement) and IS/OS disruption with non-response (Table 5) is compelling and directly applicable to clinicians. These findings deserve more emphasis in the discussion compared to the molecular data. - Figures
Figures 1 and 2 (amplification plots and melting curves) have an unacceptably low resolution and are not legible in their current form. Please provide high-quality, clearly labelled versions so readers can interpret the data. - Writing clarity
The manuscript is sometimes repetitive and grammatically inconsistent. For instance, the Abstract repeats “we want to individualize the treatment…” (lines 22–23, 34). Similarly, phrases like “elevated levels in blood and aqueous” (Table 10) appear often without precision. Careful editing would greatly improve readability. - Limitations
The limitations section should be expanded to include:
- The short follow-up period (only one month post-injection).
- Lack of validation in an independent cohort.
- The invasive nature of aqueous sampling.
- Potential confounding from HbA1c differences (Table 1).
The quality of English is below the standard required for publication. While the manuscript is understandable, it contains frequent grammatical errors, awkward phrasing, and repetitions that obscure the scientific message. Examples include repeated use of informal phrases such as “we want to individualize the treatment” (Abstract, lines 22–23, 34) and vague expressions like “elevated levels in blood and aqueous” (Table 10).
Round 2
Reviewer 1 Report
Comments and Suggestions for Authors
The authors have reflected all the said suggestions and comments, which made the manuscript enhanced with improved readability; Thus, I suggest for further consideration with acceptance.
Reviewer 2 Report
Comments and Suggestions for Authors
I noticed several inconsistencies in the numbers that should be clarified. In the Abstract, PAD4 and GDNF are significant predictors of non-response after ranibizumab injection. However, in the regression tables (Tables 8a-e), both markers lose significance once multivariate adjustment is applied, whereas MPO remains significant. This creates a mismatch between the conclusions in the Abstract/Discussion and the actual statistical output. A similar issue appears in Table 1: the mean BCVA before treatment is given as 0.98 ± 0.13 in patients and 0.84 ± 0.20 in controls, with a reported t-value of 1.406, but the P-value is shown as 0.728; these values do not align mathematically and suggest either a miscalculation or a misprint. In Table 4, HbA1c is reported as higher in non-responders (8.36 ± 0.95) compared with responders (7.76 ± 0.61), yet the difference is described as borderline non-significant (P = 0.052). This difference would likely reach statistical significance with the given means, SDs and sample sizes, so the reporting should be double-checked.
There are also a few technical details that might confuse readers. For instance, the Methods section describes ranibizumab vials as “2.3 mg in 0.23 mL,” from which 0.05 mL (0.5 mg) was injected. While the administered dose is correct, this is not how the commercial preparation is described, which could confuse. In the ROC analysis (Table 9), the combined ELANE, PAD4 and MPO panel is followed by the number “0.282,” which seems out of place in the sensitivity/specificity reporting context. Several tables also contain typographical issues, such as reversed or illogical ranges (RCVRN 1 aqueous 0.82–0.76), and stray numbers in the regression output (“0.0.733” instead of 0.733). Lastly, there are multiple minor formatting errors in the text; for example, corrupted affiliations on the title page (“opulation, 32111, EgyptPealth and …”), inconsistent table references (“Table (10 )”), and misplaced running title text.
Comments on the Quality of English Language
The English in the manuscript is understandable but not polished, with frequent long, run-on sentences, awkward phrasing, and inconsistent word use (e.g. “edema” vs. “oedema,” “non-responders” vs. “non responders”). Some expressions are informal (“changing the mind of Ophthalmologists”), and typographical errors appear on the title page and in the tables. While the scientific meaning is clear, the overall readability is reduced, and the text would benefit from careful language editing.
